# Response to Vaccines in Patients with Immune-Mediated Inflammatory Diseases: A Narrative Review

**DOI:** 10.3390/vaccines10020297

**Published:** 2022-02-15

**Authors:** Beatriz Garcillán, Miguel Salavert, José R. Regueiro, Sabela Díaz-Castroverde

**Affiliations:** 1Medical Affairs Department, Janssen, 28042 Madrid, Spain; sdiazcas@its.jnj.com; 2Infectious Disease Unit, Department of Clinical Medicine, La Fe Health Research Institute, Hospital Universitari i Politècnic La Fe, 46026 Valencia, Spain; salavert_mig@gva.es; 3Department of Immunology, Ophthalmology and ENT, School of Medicine, Complutense University, 12 de Octubre Health Research Institute (imas12), 28040 Madrid, Spain; regueiro@med.ucm.es

**Keywords:** vaccine, immune-mediated inflammatory diseases, immune response, interleukins

## Abstract

Patients with immune-mediated inflammatory diseases (IMIDs), such as rheumatoid arthritis and inflammatory bowel disease, are at increased risk of infection. International guidelines recommend vaccination to limit this risk of infection, although live attenuated vaccines are contraindicated once immunosuppressive therapy has begun. Biologic therapies used to treat IMIDs target the immune system to stop chronic pathogenic process but may also attenuate the protective immune response to vaccines. Here, we review the current knowledge regarding vaccine responses in IMID patients receiving treatment with biologic therapies, with a focus on the interleukin (IL)-12/23 inhibitors. B cell-depleting therapies, such as rituximab, strongly impair vaccines immunogenicity, and tumor necrosis factor (TNF) inhibitors and the cytotoxic T-lymphocyte-associated antigen-4 (CTLA-4) fusion protein abatacept are also associated with attenuated antibody responses, which are further diminished in patients taking concomitant immunosuppressants. On the other hand, integrin, IL-6, IL-12/23, IL-17, and B-cell activating factor (BAFF) inhibitors do not appear to affect the immune response to several vaccines evaluated. Importantly, treatment with biologic therapies in IMID patients is not associated with an increased risk of infection with severe acute respiratory syndrome coronavirus 2 (SARS-CoV-2) or developing severe disease. However, the efficacy of SARS-CoV-2 vaccines on IMID patients may be reduced compared with healthy individuals. The impact of biologic therapies on the response to SARS-CoV-2 vaccines seems to replicate what has been described for other vaccines. SARS-CoV-2 vaccination appears to be safe and well tolerated in IMID patients. Attenuated but, in general, still protective responses to SARS-CoV-2 vaccination in the context of certain therapies warrant current recommendations for a third primary dose in IMID patients treated with immunosuppressive drugs.

## 1. Introduction

Immune-mediated inflammatory diseases (IMIDs) are a diverse group of clinically unrelated conditions, including, among others, rheumatoid arthritis (RA), psoriatic arthritis (PsA), psoriasis (PsO), and inflammatory bowel disease (IBD). They share molecular mechanisms and are characterized by altered immune homeostasis that results in chronic excessive inflammation leading to tissue injury in affected organs and eventually functional disability [1].

Due to their inflammatory nature, IMIDs are often treated with immunosuppressants or immunomodulators, including steroids, methotrexate (MTX), thiopurines, small-molecule inhibitors, and a range of biologic therapies that target key molecules or cells involved in the inflammatory response. Patients with IMIDs are often at increased risk of infections because of the disease itself, the immunosuppressive or immunomodulators used to treat the disease, comorbidities, and/or hospitalizations caused by disease flares or complications [2,3,4]. Infections can be prevented through the use of chemoprophylaxis and/or vaccination; in fact, guidelines for the management of IMID patients recommend vaccination in accordance with local immunization schedules and patient-specific risks, unless there are contraindications [5,6,7,8,9].

The current severe acute respiratory syndrome coronavirus 2 (SARS-CoV-2) pandemic and the advent of population-wide vaccination programs raise questions about the safety and efficacy of SARS-CoV-2 vaccination in IMID patients, especially those under immunomodulatory or immunosuppressive therapy. A range of SARS-CoV-2 vaccines are available or in development. Most employ non-replicating viral vectors or mRNA for the spike protein of the virus, but some use more conventional technology, such as inactivated virus or live attenuated virus [10,11]. It is likely that the vaccine’s ability to generate an effective immune response against SARS-CoV-2 will be affected by the type of medications patients are receiving. The aim of this narrative review is to summarize current knowledge regarding vaccine responses in IMID patients treated with biologic therapies, with a focus on agents that inhibit interleukin (IL)-12 and/or -23, as various new IL-23 inhibitors are being approved for several IMID conditions.

## 2. Increased Risk of Infections in Patients with IMIDs

Patients with IMIDs have an increased risk of infections compared with unaffected individuals, firstly, because of the immune dysfunction that characterizes these conditions and, secondly, due to the immune modulatory therapies used to treat them. Corticosteroids are associated with the highest risk of infections [12,13], and thiopurines and MTX also increase the risk of infections compared with placebo [14,15,16], but the magnitude of the risk increase is not as high as with corticosteroids.

Some biologic therapies used to treat IMIDs may also increase the risk of infections, but the risk is not uniform across all agents. The European Society of Clinical Microbiology and Infectious Disease (ESCMID) reviewed the evidence for biologic therapies and concluded that TNF inhibitors increase the risk of active tuberculosis and other granulomatous infections and may also increase the risk of other serious bacterial, fungal, opportunistic, and certain viral infections [17]. The infection risk associated with biologic therapies targeting IL-6 or the IL-6 receptor is similar to the risk with TNF inhibitors [18]. IL-1 inhibitors are associated with a moderate increase in the risk of infections, which are generally mild to moderate in most patients [18]. IL-17 inhibitors are associated with an increased risk of Candida infections [19]. IL-12/23 inhibitor ustekinumab is not associated with an appreciable increase in the risk of infections [18]. Anti-integrins do not appear to increase the risk of infections, except for anti-α4 integrins that increase the risk of progressive multifocal leukoencephalopathy (PML) by Polyomavirus JC [20]. Agents targeting CD20 (e.g., rituximab) increase the risk of infections, the most common being respiratory tract infections, hepatitis B (and, to a lesser extent, hepatitis C) reactivation, and varicella zoster infection [21].

Current data suggest that patients with IMIDs are not at increased risk of acquiring SARS-CoV-2 infection or of hospitalization/death compared with the general population [22,23,24,25]. However, older age, comorbidities, and the use of non-biologic systemic therapy (particularly corticosteroids) increases the risk of severe COVID-19 disease and/or hospitalization in patients with IBD, PsO, or rheumatic IMIDs [26,27,28,29,30,31]. Biologic therapies (except for the B-cell depleting agent rituximab [32]) were not associated with an increased risk of developing severe COVID-19 [33,34], and some of these therapies may in fact be protective against COVID-19-related hospitalization and death among infected patients [26,27,28,29,35,36].

Based on the increased risk of infection, preventive measures are an important strategy for limiting serious illness in patients with IMIDs, either by appropriate vaccination or by chemoprophylaxis to prevent reactivation of latent infections, such as tuberculosis, Pneumocystis, or hepatitis B.

## 3. Vaccines and IMIDs

Multiple sets of international guidelines recommend vaccination to limit the risk of infection in patients with IMIDs [5,6,7,8,9]. Before initiating systemic immunosuppressive therapy, a vaccination and infectious disease history should be undertaken to identify gaps in the patient’s vaccination schedule, and vaccinations should be administered according to the recommendations for the general population and for the specific patient group.

Ideally, patients should be vaccinated before immunosuppressive therapy is started, as therapies could impact vaccine efficacy. Moreover, live attenuated vaccines could cause disease in immunocompromised patients and are therefore contraindicated once immunosuppressive therapy has been initiated, unless the potential benefit of preventing the infection outweighs the risk of administering a live vaccine [37]. Inactivated vaccines are generally safe and do not increase the risk of IMID exacerbation, although some studies reported increased joint pain following the administration of some vaccines in patients with rheumatologic conditions [38,39].

Regular influenza and pneumococcal vaccinations are deemed particularly important in IMID patients on biologic therapy. Physicians may also consider vaccination against Haemophilus influenzae type b, Neisseria meningitidis, hepatitis B, diphtheria, and tetanus, depending on the patient’s vaccination history and clinical profile [6,7,8,40].

Despite the importance of vaccination in IMID patients, suboptimal vaccination rates are reported. Possible reasons include lack of awareness of recommendations, inadequate communication between specialist and primary care physicians, and concerns about the efficacy and safety of vaccines [7,41,42].

## 4. Immune Response Necessary to Generate Immunity

Vaccines are one of the key developments of modern medicine; they prevent thousands of infections and deaths worldwide and represent an important advance in public health.

Vaccination provides protective immunity by eliciting a pathogen-specific immune response against the antigenic material present in the vaccine; this immune response is often boosted by adjuvants within the vaccine that enhance the magnitude and durability of the response [43]. The main immune effectors of vaccine responses are antibodies (humoral immunity) and T cells (cellular immunity). Antigen(s) are taken up by antigen-presenting cells (APCs) and are presented to B and T cells to generate antigen-specific B- and T-cell responses [44]. Antibodies may prevent or reduce infection by binding to the pathogen, while T cells mediate elimination of infected cells. CD4+ helper T cells (Th) orchestrate the immune response by providing T-cell help to primed B cells and CD8+ cytotoxic T cells. In addition, Th cells secrete cytokines that aid pathogen clearance. In lymphoid follicles, T follicular helper cells (Tfh cells) provide T-cell help to B cells and mediate their activation and differentiation into antibody-producing plasma cells and memory B cells.

Once an infection is cleared, most of the antigen-specific cells die off through apoptosis, but antibodies and long-lived plasma cells persist along with memory B and T cells, which can mount an anamnestic response upon re-exposure to the antigen [45].

In most cases, to induce an antibody response, B cells need the aid of Tfh cells (T cell-dependent B-cell response). However, non-protein antigens, such as pure polysaccharide vaccines, activate B cells and elicit antibody production without the involvement of T cells (T cell-independent B-cell response). This T-independent response is generally considered to produce low affinity IgM antibodies and to be short-lived [44].

Vaccine-induced immune responses are often measured as a correlate of vaccine efficacy; most commonly, the humoral immune response is measured by quantifying antigen-specific antibodies in plasma (e.g., total IgG antibodies measured by ELISA) or, if feasible, functional antibodies (e.g., neutralizing antibodies). Some studies report seroprotection and/or seroconversion rates; seroprotection is defined as the antibody level needed to achieve protection, while seroconversion refers to a fold-rise in antibody levels from pre- to post-vaccination. The measurement of cellular immunity is less common and is based in the detection of antigen-specific CD4+ and CD8+ T cells and the cytokines they release following in vitro stimulation.

A key goal of vaccination is to induce long-lasting immunity; however, while some vaccine responses are durable (e.g., hepatitis A, B, and diphtheria), others are short-lived and require multiple administrations (e.g., influenza). The magnitude and duration of vaccine-induced immunity is determined by host, environmental and vaccine factors (structure of the antigen; adjuvants; dose; schedule, site, route, and timing of administration; and concomitant medications) [45,46].

## 5. Approved Biologic Therapies and Their Impact on Responses to Vaccines

Immunosuppressive/immunomodulatory drugs used to treat IMIDs target the immune system to stop pathogenic chronic inflammation, but, in many cases, these therapies may also impact the ability to properly respond to infections and to vaccines. A range of biologic therapies are used in IMIDs; they act at different stages of the inflammatory response, by depleting certain immune cells, or by blocking their activation or their migration. Below, we review the impact of biologic therapies on vaccine immunogenicity in IMID patients; the evidence is also summarized in Table 1 and Table 2.

### 5.1. Integrins

#### 5.1.1. Anti-α4 (Natalizumab)

Integrin antagonists inhibit the trafficking of circulating immune cells to the site of inflammation. Natalizumab, a monoclonal antibody (mAb) against the α4 chain common to the α4β1 and α4β7 integrins, blocks the migration of leukocytes to the central nervous system and gastrointestinal (GI) tract by blocking the interaction between α4β1 and vascular cell adhesion molecule 1 (VCAM-1) and between α4β7 and mucosal addressin cell adhesion molecule 1 (MAdCAM-1), respectively. It is widely approved for the treatment of multiple sclerosis (MS) but carries a risk of PML. In the US, natalizumab is also available through a restricted access program for patients with moderate to severe active Crohn’s disease (CD) who have an inadequate response or cannot tolerate other therapeutic options.

Data on the effect of natalizumab on vaccine response are scarce (Table 1). Two small studies showed a reduced rate of seroconversion after seasonal influenza vaccination in MS patients compared with healthy controls [47,48]. Additionally, in a randomized open-label study in MS patients treated or not with natalizumab, all patients achieved protective levels of antibodies to a recall antigen (tetanus toxoid) or a neoantigen (keyhole limpet hemocyanin [KLH]—a carrier protein used to research antibody production) [49].

#### 5.1.2. Anti-α4β7 (Vedolizumab)

Vedolizumab inhibits the extravasation of leukocytes to the intestinal mucosa by selectively blocking the integrin α4β7, which targets MAdCAM-1, a cell adhesion molecule that is expressed by endothelial cells in the GI tract [108]. Vedolizumab is indicated for the treatment of CD and ulcerative colitis (UC).

The effect of vedolizumab on vaccine responses was evaluated in a phase 1, randomized, double-blind, placebo-controlled trial. Healthy individuals received a single intravenous dose of vedolizumab of 750 mg (approved dose is 300 mg) (Table 1). Vedolizumab did not alter the response to a parenterally administered vaccine (hepatitis B), as antibody titers were similar between placebo and vedolizumab groups. However, the humoral response to an enterally administered vaccine (oral cholera vaccine) was significantly reduced, probably due to the gut-specific effect of this agent [51]. A recent study analyzed the immunogenicity of the trivalent or quadrivalent influenza vaccine in 19 patients with IBD who were receiving vedolizumab. The humoral response (antibody titers, seroprotection, and seroconversion rates) to seasonal influenza vaccine was similar in patients on vedolizumab and in the 20 healthy controls [50].

### 5.2. Co-Stimulators of T Cells

CTLA-4 Fusion Protein (Abatacept)

Abatacept is a soluble fusion protein comprising the extracellular domain of human cytotoxic T-lymphocyte-associated antigen-4 (CTLA-4). It acts by blocking T-cell activation, as it competes with CD28 for binding to CD80/86 on APCs, a required second co-stimulatory signal needed for T-cell activation [109]. Abatacept is indicated for the treatment of RA, PsA, and polyarticular juvenile idiopathic arthritis (JIA).

Vaccination effectiveness while on abatacept therapy has been assessed only in RA patients (Table 1). The response to the 23-valent polysaccharide pneumococcal vaccine (PPV-23), which induces a T cell-independent response, was analyzed in a randomized, double-blind, controlled trial, and a diminished IgG response was observed with abatacept. However, in a functional assay (opsonization index), no differences were found between groups (abatacept, MTX, or control) [52]. Moreover, in two open-label sub-studies with no control group, RA patients receiving abatacept mounted a proper immune response to PPV-23 and influenza vaccine [53]. However, the antibody response to the pneumococcal conjugate vaccine (PCV), which induces a T cell-dependent response, was shown to be impaired in a small cohort of RA patients receiving abatacept (mostly in combination with MTX) [54]. Similarly, abatacept treatment was associated with a significantly reduced IgG response to the monovalent A/H1N1 influenza vaccine compared with MTX treatment [55].

In addition to studies evaluating vaccine response during treatment with abatacept, a study in children aged 2–5 years found that starting abatacept treatment did not affect pre-existing protective antibody levels to tetanus and diphtheria vaccinations administered 21–79 months before abatacept initiation [110].

### 5.3. B Cells

B cells are essential for humoral immune responses but have also been implicated as drivers of several autoimmune and IMID conditions. B-cell targeted biologic therapies are now available for the treatment of IMIDs [111]. These include the CD20-specific B-cell depleting therapies, such as rituximab, and the B-cell activating factor (BAFF) inhibitor belimumab [112,113].

#### 5.3.1. Anti-CD20 (Rituximab)

Rituximab is a mAb that binds to the B-cell surface protein CD20 and leads to the depletion of B lymphocytes. It is indicated for RA, MS, and B-cell related hematologic malignancies and has been investigated in the treatment of systemic lupus erythematosus (SLE) [111,114].

Since B cells are responsible for humoral immune responses, it is unsurprising that patients receiving rituximab show reduced antibody levels after vaccination. Bingham and colleagues found that, compared with RA patients taking MTX alone, a lower proportion of patients taking rituximab + MTX mounted a protective antibody response to the T-cell independent vaccine PPV-23. In contrast, a similar proportion responded to the T-cell dependent tetanus toxoid vaccine [56].

The magnitude of the antibody response to PCV, and the proportion of patients mounting a protective antibody response, was also reduced in patients with RA receiving rituximab monotherapy and in those receiving rituximab + MTX compared with those receiving MTX monotherapy [54]. Similarly, in the Red de Investigación en Inflamación y Enfermedades Reumáticas (RIER) study, which investigated the response to vaccines among patients receiving biologics for various IMIDs (arthropathies, connective tissue diseases, PsO, or IBD), functional antibody levels were lower after pneumococcal vaccination with PCV-7 and PPV-23 in patients receiving rituximab than in those receiving other biologic therapies [57]. The same study also reported that a lower proportion of patients under rituximab developed protective antibody titers after hepatitis B vaccination compared with patients receiving other types of biologics [58].

Response to the trivalent influenza vaccine in patients currently receiving or recently completing treatment with rituximab has been examined in several studies. Patients with RA who had received rituximab showed an impaired humoral response compared with RA patients receiving non-biologic disease modifying anti-rheumatic drugs (DMARDs) or healthy controls [59,60,61,62,63]. However, the cellular immune response to influenza vaccination was not affected by rituximab, with similar levels of influenza-specific CD4+ cells in patients treated with rituximab or DMARDs [59]. Interestingly, antibody response might be improved by a longer dose-administration schedule between rituximab administration and influenza vaccination [61,63].

Similar to responses to the trivalent influenza, a diminished antibody response to the monovalent H1N1 influenza vaccine has been reported in patients taking rituximab [64,65]. Adler and colleagues examined the response to H1N1 vaccination in IMID patients, of whom eight were receiving rituximab and none of them developed seroprotective antibody levels [65]. In a study from Sweden, RA patients receiving rituximab had a significantly reduced humoral response compared with those receiving MTX [64].

Due to the impact of rituximab treatment on vaccine immunogenicity, several guidelines have made recommendations regarding timing of vaccination while on rituximab. IMID patients normally receive rituximab cycles every 6 months and the general recommendation is to administer inactivated vaccines at least 5 months following the last rituximab dose to allow for some B-cell reconstitution [115,116].

#### 5.3.2. Anti-BAFF (Belimumab)

The BAFF inhibitor belimumab acts by neutralizing BAFF (also known as B lymphocyte stimulator), a key cytokine involved in B-cell survival and differentiation [114,117]. Belimumab is indicated for SLE, the first biologic approved for this condition [113]. The B cell-depleting effect of belimumab is not as intense as that of rituximab, since belimumab preferentially targets transitional and naïve B cells while leaving conventional memory B cells unaffected [118]; therefore, the two different classes of B cell-depleting therapies may have different effects on the vaccine response [114]. Nevertheless, because it targets B cells, belimumab may negatively impact the response to vaccines.

Currently, data on the effects of belimumab are limited (Table 1), but studies have shown that SLE patients receiving belimumab who were immunized with PCV generated a similar antibody response to SLE patients receiving other standard immunosuppressive therapies (DMARDs, azathioprine, MTX, and/or corticosteroids) [66].

Moreover, in a phase 4, open-label study, immunization of SLE patients with PPV-23 4 weeks before or 24 weeks after starting belimumab resulted in similar humoral response [67].

Consistent with preservation of memory B cells by belimumab, pre-existing antibody titers to pneumococcus, tetanus, or influenza vaccinations received before starting belimumab were shown to be preserved in a sub-study from the phase 3 randomized placebo-controlled trial [119].

### 5.4. Pro-Inflammatory Cytokines

#### 5.4.1. Anti-TNF

TNF is a proinflammatory cytokine that is important in immune responses, particularly to intracellular pathogens. However, dysregulated TNF production leads to excessive inflammation and survival of pathogenic autoreactive immune cells and, as such, TNF plays a central pathophysiologic role in a range of IMIDs [120,121].

Agents targeting TNF are widely used in the treatment of PsA, PsO, RA, spondyloarthropathies, and IBD and include infliximab, etanercept, adalimumab, golimumab, and certolizumab pegol. Due to the central role of TNF in immunity and the widespread use of TNF inhibitors, the impact of anti-TNF therapy on vaccine responses has been extensively studied (Table 1).

Since some of the proinflammatory actions of TNF include maturation of APCs, costimulation of T cells, induction of the germinal center (GC), and stimulation of immunoglobulin synthesis [68], it is possible that inhibition of TNF may impair vaccine responses.

The antibody response to PPV-23 appears to be moderately affected by TNF inhibitor treatment, although the data are affected by the choice of comparator and by concomitant medications. Studies in patients with rheumatological IMIDs have shown that, compared with patients taking placebo or NSAIDs, those receiving TNF inhibitors generate a less robust antibody response to PPV-23 [68,71,74,75,78]. However, other studies showed no difference in antibody titer or the proportion of patients achieving protective antibody levels after PPV-23 between RA patients receiving TNF inhibitors and those receiving placebo [69,72]. Studies comparing the vaccine response in patients receiving TNF inhibitors and those receiving MTX have generally shown no significant difference between the two groups of patients [69,70,73,122], but MTX itself blunts the humoral response to PPV-23 [123,124]. The effect of TNF inhibitors and MTX administered together appears to be additive, since the most marked effect of TNF inhibitors on the antibody response to PPV-23 has been shown in patients taking concomitant MTX [69,72]. Most of these studies were undertaken in patients with rheumatological IMIDs, but studies in patients with IBD also showed lower antibody titers after PPV-23 in patients receiving TNF inhibitors ± immunosuppressive therapy (azathioprine) than in those receiving mesalamine or immunosuppressive monotherapy [74,75,76,77].

Impaired humoral responses to PCV vaccination were also seen in patients with RA, spondyloarthropathies, or IBD taking TNF inhibitors [39,74,78], particularly in the presence of concomitant MTX [39].

Lower rates of seroconversion to the hepatitis B vaccine have been reported during TNF inhibitor therapy in patients with spondyloarthropathies [68] or IBD [79,80], although another study found no difference in seroconversion rates between IBD patients receiving TNF inhibitors, or immunomodulators, or their combination [81]. The seroprotection rate after hepatitis B vaccine tends to be diminished in patients with IBD [81], but this is especially marked among IBD patients receiving TNF inhibitors [80]. Some authors suggest that IBD patients receiving TNF inhibitors should receive two full courses of hepatitis B vaccine, instead of one, to achieve protection [80]. Similarly, many patients receiving TNF inhibitors and/or MTX do not achieve seroprotection after a single dose of hepatitis A virus (HAV) vaccine [82].

A diminished antibody response to seasonal influenza vaccine in patients receiving TNF inhibitors has been reported in several studies [83,84,85,89,90,93,94]; however, some other studies have reported a normal humoral response [86,87,88]. The timing of the antibody assessment in these studies may explain this discrepancy, for example, a longitudinal study in IBD patients showed a preserved antibody response at 3 weeks but a lower antibody titer at 6 months and 2 years in TNF inhibitor recipients vs. controls [91]. As with the PPV-23 vaccine, the response to influenza vaccination may be further blunted in patients who take concomitant immunosuppressants in combination with the TNF inhibitor [89,90,92,125,126].

Live attenuated vaccines are contraindicated in patients under anti-TNF therapy; however, some small studies have reported some cases of exposure. The humoral and cellular response to measles-mumps-rubella (MMR) was preserved in 5 children with JIA on low-dose MTX in combination with etanercept [95]. Moreover, in a Brazilian case series of 17 RA patients under infliximab therapy, yellow fever revaccination resulted in an adequate humoral response [96]. Importantly, these two studies did not report adverse effects or secondary severe infections after vaccination with live attenuated vaccines. Similarly, a study evaluating the safety of the live attenuated herpes zoster (HZ) vaccine in 551 IMID patients under anti-TNF therapy did not find cases of varicella or HZ 42 days after vaccination [127]. Despite these encouraging results, larger studies are needed to evaluate the risks with this type of vaccines.

Despite a somewhat attenuated antibody response to vaccines in patients receiving TNF inhibitors, a high proportion of patients nevertheless achieve protective antibody levels, supporting current recommendations to vaccinate these patients.

#### 5.4.2. Anti-IL-1

IL-1 is a family of 11 cytokines involved in the innate immune response, with both pro- and anti-inflammatory functions. IL-1β and IL-1α, specifically, have pro-inflammatory properties and are involved in the development of rheumatic diseases [128].

Anakinra, a recombinant IL-1 receptor antagonist, blocks the interaction of IL-1β and IL-1α with its receptor and is indicated for RA and cryopyrin-associated periodic syndromes (CAPS). Canakinumab, an anti-IL-1β mAb, is approved for periodic fever syndromes (such as CAPS) and systemic JIA.

The RIER study included only one patient taking anakinra in their analyses of the response to influenza or hepatitis B vaccines; therefore, no meaningful assessment of the effect of anakinra on vaccine response was possible [58,129].

To date, vaccine responses in patients under canakinumab therapy have been studied in children with CAPS. In an open-label phase 3 study, canakinumab had no effect on antibody titers after immunization with non-live childhood vaccines [130].

#### 5.4.3. Anti-IL-17

IL-17 cytokines are produced by type 17 cells, which include Th17 cells, subsets of γδ T cells, invariant NK T cells, ‘natural’ Th17 cells, type 3 innate lymphoid cells (ILCs), and mucosal-associated invariant T (MAIT) cells [131]. IL-17 plays a role in protective immunity against extracellular bacterial and fungal infections [132]. IL-17-producing cells accumulate at mucosal surfaces where they are responsible for the maintenance of barrier integrity, production of antimicrobial peptides, and recruitment of neutrophils. Increased numbers of IL-17-producing cells have been found in the skin of patients with PsO and the joints of patients with RA [133,134], and increased levels of type 3 ILCs have been found in the synovia and blood of patients with PsA, which correlate with disease activity [135]. In CD, increased expression of IL-17A has been reported in the intestinal mucosa [136]. In spite of that, IL-17 inhibitors have proven to be ineffective in patients with IBD, and cases of de novo IBD or IBD exacerbation have been reported [137].

IL-17 inhibitors include the anti-IL-17A mAbs ixekizumab and secukinumab and the IL-17RA mAb brodalumab [113]. Secukinumab and ixekizumab are approved for plaque PsO, PsA, and axial spondyloarthropathies, whereas brodalumab is only approved for PsO.

The impact of IL-17 inhibition on vaccine responses has only been assessed in three small studies (Table 1). In an open-label randomized study, healthy subjects received a single 150 mg dose (approved dose 300 mg) of secukinumab or no treatment 2 weeks prior to vaccination with influenza and group C meningococcal vaccine. Seroconversion rates 4 weeks later were similar between the two groups [97]. Similarly, a separate study showed that the humoral response to the seasonal influenza vaccine in patients with PsA or ankylosing spondylitis receiving secukinumab was similar to the response observed in healthy controls [98]. In a randomized study in healthy adults, administration of two doses of ixekizumab (2 weeks before and on the day of tetanus and PPV-23 vaccination) did not significantly reduce the seroprotection rates when compared with individuals not receiving ixekizumab [99].

#### 5.4.4. Anti-IL-6

IL-6 is a master cytokine that regulates innate and adaptive immune responses. IL-6 is secreted during infection or inflammation by many cell types, such as endothelial and synovial cells in joints [138,139]. Dysregulation of IL-6 signaling can cause persistent synovial inflammation and damage to the articular cartilage and underlying bone and eventually leads to the development of inflammatory arthritis [140]. Currently available IL-6 receptor inhibitors include tocilizumab and sarilumab, both of which are approved for RA; tocilizumab is also approved for systemic JIA.

Vaccine efficacy in patients receiving IL-6 inhibitors might be affected for several reasons: IL-6 plays an important role in naïve T cell differentiation, it is a growth factor for B cells, and promotes the development of Tfh cells [139]. However, studies investigating the impact of tocilizumab in response to PPV-23 (Table 1) suggest that tocilizumab did not significantly reduce the antibody response compared with patients receiving other treatments (MTX, anti-TNF, or conventional DMARDs) [100,101,102], Similarly, the humoral response to the conjugate pneumococcal vaccine was similar in RA patients receiving tocilizumab and those not receiving biologic therapy [54]. Tocilizumab does not appear to affect the antibody response to influenza vaccination, including in children with JIA [64,102,103,104], but concomitant treatment with MTX does abrogate the antibody response in tocilizumab recipients [102,103]. Lastly, in a single randomized study, tocilizumab had no effect on the antibody response to tetanus toxoid vaccine [100].

## 6. Vaccine Response in Patients Treated with IL12/23 and IL-23 Inhibitors

IL-12 and IL-23 are produced by innate immune cells, mainly macrophages and dendritic cells, and are an important link between innate and adaptive immunity. They are structurally similar but exert different functions in immunity against pathogens, as well as in the pathogenesis of disease. IL-12 and IL-23 are heterodimeric cytokines; IL-12 is composed of the p40 and p35 subunits, while IL-23 is formed by p40 and p19. In addition to sharing the p40 subunit, IL-12 and IL-23 also share receptor chains. Their corresponding receptors are heterodimeric complexes, the receptor for IL-12 consists of IL-12Rβ1 and of IL-12Rβ2, whereas the receptor for IL-23 is made up of IL-12Rβ1 and IL-23R (Figure 1) [141].

In 1989, IL-12 was identified as a potent growth factor and activator of NK cell functions, including interferon-γ (IFN-γ) production and cytotoxic activity [142]. Subsequently, IL-12 was found to drive the differentiation of naïve CD4+ T cells into Th1 cells, which orchestrate adaptive immune responses [143]. Binding of IL-12 to its receptor leads to IFN-γ production, which plays a critical role in host defense against intracellular pathogens [144].

For many years, IL-12-mediated Th1 responses were thought to be responsible for several autoimmune and autoinflammatory conditions. This notion was based on experimental animal models using mice deficient in p40 or IL-12Rβ1 and suggested that IL-12 could be an interesting therapeutic target. The discovery of IL-23 in 2000 [145] led to a paradigm-changing study by Cua and colleagues in 2003, in which the authors described that IL-23 and not IL-12 (as previously thought) was the culprit for the initiation of autoinflammation in a murine model of experimental autoimmune encephalomyelitis [146].

IL-23 stimulates type-17 cells to produce IL-17A, IL-17F, and IL-22 and is also responsible for the maintenance and expansion of Th17 cells, which have been implicated in chronic inflammation and are thought to be the drivers of several IMIDs. In fact, single nucleotide polymorphisms in IL-23R have been linked with the development of several IMIDs [147], highlighting the role of the IL-23/Th17 axis in the pathogenesis of IMIDs.

Several mAbs have been developed to target IL-12 and/or IL-23. Ustekinumab, a human mAb against the p40 subunit targets both IL-12 and IL-23, is approved for the treatment of PsO, PsA, CD, and UC. Appreciation of the predominant role of the IL-23/Th17 axis in the pathogenesis of IMIDs led to the development of antibodies against the p19 subunit, which exclusively target IL-23. Guselkumab, tildrakizumab, and risankizumab are all approved for the treatment of PsO; guselkumab and risankizumab are also approved for PsA.

More recently, IL-12 and IL-23 have also been implicated in the differentiation of CD4+ Tfh cells, which provide B-cell help in the GC to generate high-affinity antibodies and to promote the differentiation of B cells into memory B cells or long-lived plasma cells [148,149]. Owing to their functions, Tfh cells are key to immune responses to pathogens and vaccines and may be potentially impacted by IL-12/23 inhibition.

To date, there are limited clinical data on the effects of IL-12/23 inhibition on vaccine responses, but available data suggest that a sufficient vaccine response is likely in patients taking ustekinumab and, by extension, IL-23-specific inhibitors. In addition, data from individuals with complete genetic deficiency of these cytokines or their receptors may shed some light on the impact of IL-12/23 inhibition on responses to pathogens and vaccines.

Monogenic inborn errors leading to complete deficiency of IL-12 and/or IL-23 or their receptors are often characterized by an increased susceptibility to infections with poorly pathogenic mycobacteria and salmonella due to impaired IFN-γ production in otherwise healthy individuals. However, penetrance is not complete, and the phenotype differs between individuals [150,151,152].

Due to the role of IL-12/23 in Tfh cell commitment, it was hypothesized that its deficiency would impact antibody responses. Subjects lacking IL-12Rβ1 (abolished IL-12 and IL-23 signaling) are still able to generate Tfh cells [148] and are not prone to B cell-deficient associated infections [153]. Moreover, they show normal serum specific IgG titers for tetanus toxoid, rubella, Epstein-Barr virus, cytomegalovirus, and varicella virus [148] and preserved humoral and cellular immune responses to influenza vaccination [150]. These antibodies may be generated independently of the GC response, and therefore would be expected to show low avidity for their antigen. In fact, the avidities of specific IgG against the tetanus toxoid were lower in IL-12Rβ1-deficient subjects, but the avidities of rubella virus-specific IgG were comparable to healthy subjects [148].

During the early development of ustekinumab, humoral response to immunization with KLH was assessed in cynomolgus monkeys during the preclinical toxicology study. Monkeys were treated with multiple doses of ustekinumab or placebo for 26 weeks, and anti-KLH antibody titers were similar in both groups, even when using ustekinumab levels exponentially higher than those approved in humans [106].

Responses to vaccination were also assessed during phase 1 studies in patients with PsO and MS who had received a single dose of ustekinumab. Humoral recall response to the PPV-23 and tetanus toxoid vaccines were similar between patients receiving ustekinumab or placebo (Table 1) [106].

Two further studies evaluated the impact of ongoing use of ustekinumab on vaccine responses (Table 1) [105,107]. The response to tetanus toxoid and PPV-23 vaccines was assessed in a subset of PsO patients who had received ustekinumab for at least 3 years in the phase 3 PHOENIX 2 clinical trial (*n* = 60) and a control group of PsO patients not receiving systemic therapy (*n* = 56) [105]. Humoral and cellular immune responses were comparable between ustekinumab-treated and control groups. Four weeks after vaccination, seroconversion rates to the tetanus and PPV-23 vaccines were similar in the control group and in those receiving ustekinumab. Importantly, this study investigated the impact on vaccine responses of long-term treatment with ustekinumab in patients with PsO. Moreover, these data show that inhibition of IL-12/23 with ustekinumab does not compromise immune response to T-dependent (tetanus) or T-independent (PPV-23) vaccines in patients with PsO during long-term use of ustekinumab [105].

Similarly, a recent small prospective study evaluated the humoral and cellular immune response to seasonal influenza vaccine in patients with CD treated with ustekinumab and in healthy controls. Three months after vaccination, functional antibody responses were measured using hemagglutinin inhibition assays, and based on these results, seroprotection and seroconversion rates to the three influenza strains in the vaccine were calculated. Seroprotection and seroconversion rates were high and comparable between ustekinumab-treated patients and healthy controls. Importantly, this study also showed that ustekinumab did not impair cellular immune responses, as assessed by the proliferation of influenza-specific CD3+, CD4+, and CD8+ T cells [107]. The lack of effect of ustekinumab on cellular immune responses is encouraging and important since this measure of vaccine response is not always included in assessments of vaccine immunogenicity.

In conclusion, pharmacologic inhibition of IL-12/23 is probably milder than complete congenital immunodeficiency. In fact, the increased susceptibility to infections with poorly pathogenic mycobacteria and salmonella observed in immunodeficient patients has not been reported in patients treated with ustekinumab. Regarding impaired Tfh functions, data on congenital immunodeficient patients show a mild effect, whereas these functions seem to be preserved during treatment with IL-12/23 inhibitors [148,154,155].

Data are not yet available on the effects of IL-23 inhibition on vaccine responses; however, since inhibition of both IL-12 and IL-23 with ustekinumab does not seem to affect the immune response induced by several vaccines, it is likely that IL-23-specific inhibition will similarly show minimal impact on protective immunity after vaccination.

## 7. SARS-CoV-2 Vaccination in IMID Patients

Given the scale of the global COVID-19 pandemic, it is crucial that vaccination provides effective protection against SARS-CoV-2 infection. The current vaccines, which were rapidly developed and, in some cases, incorporate novel technologies, have proven to be effective in healthy individuals. However, patients receiving immunosuppressive therapies were excluded from the phase 3 trials, raising questions about the effectiveness and safety of the vaccines in these patients. All available SARS-CoV-2 vaccines, as well as almost all in the development pipeline, are non-live vaccines that employ non-replicating viral vectors or mRNA [156], so theoretically they are not contraindicated in patients with IMIDs receiving systemic immunosuppressive or immunomodulatory therapies.

At the time of writing this review, several reports are being published on the immunogenicity and safety of SARS-CoV-2 vaccines in IMID patients and the impact of biologic therapy (summarized in Table 3).

### 7.1. Humoral Responses to SARS-CoV-2 in IMIDs

The COVaRiPAD study in the US is a prospective, observational study to assess the immunogenicity of mRNA vaccines in 133 patients with chronic inflammatory diseases (CID); the most common conditions were IBD and RA. One to two weeks after the second dose of vaccine, CID patients showed a reduced humoral response compared with healthy controls. Regarding immunosuppressive medications, corticosteroids and B cell-depleting therapies substantially affected vaccine immunogenicity, with 36- and 10-fold reductions in humoral response, respectively. Janus kinase (JAK) inhibitors and antimetabolites (e.g., MTX) were also associated with attenuated antibody and neutralization titers, whereas TNF, IL-12/23, and integrin inhibitors had minimal impact on humoral response [157].

Similarly, two German studies reported that patients with CID or IMIDs may develop a less robust humoral response compared with healthy individuals; however, most patients will develop a sufficient response to be considered protected [158,159]. Simon and colleagues showed lower titers of anti-SARS-CoV-2 IgG antibodies in 84 IMID patients compared with healthy controls after vaccination with the BNT162b2 (Pfizer-BioNTech) mRNA vaccine. They found that the impaired immune response was not associated with any individual immunomodulatory treatment but rather the disease itself [158]. Similarly, Geisen and colleagues found a lower anti-SARS-CoV-2 antibody titer 7 days after vaccination with mRNA vaccines in a small cohort of CID patients (*n* = 26) than in healthy controls, but all the vaccinated individuals developed a protective level of neutralizing antibodies [159].

Data from Israeli patients with inflammatory rheumatic diseases showed a significant humoral response (antibody concentration >50 U/mL) in most patients (86%) following two doses of BNT162b2 (Pfizer-BioNTech) mRNA vaccine, although those receiving rituximab showed significantly impaired humoral response, particularly in older patients [160].

A Swiss study evaluated the humoral response to mRNA vaccines in 53 RA patients receiving DMARDs and found that anti-SARS-CoV-2 titers were significantly lower in RA patients compared with healthy volunteers. In fact, after one dose, only 10% of RA patients versus 90% of controls developed antibody levels associated with protection (>15 U/mL). After the second dose, 88% of RA patients and 100% of controls achieved protective antibody levels. The lowest response was found in those patients receiving JAK inhibitors, with only 67% reaching protective levels [161].

Researchers from the University Hospital of Pavia, Italy, analyzed the immunogenicity of a single dose of the BNT162b2 (Pfizer-BioNTech) mRNA vaccine in patients with rheumatologic IMIDs. Among patients not receiving glucocorticoids or MTX, the rate of response (antibody concentration > 15 U/mL) was 85.4%, similar to that reported in registration trials, whereas patients receiving both glucocorticoids and MTX showed a decreased seroconversion rate (33%). In contrast, anti-cytokine therapy showed no impact on immunogenicity [162]. Similarly, two monocentric studies showed reduced vaccine immunogenicity in IMID patients treated with MTX compared with biologics or non-MTX oral medications [163]. In contrast, Venerito and colleagues found preserved humoral responses to BNT162b2 (Pfizer-BioNTech) mRNA vaccine in PsA patients irrespective of the immunomodifying therapy (TNF inhibitors, MTX, glucocorticoids, or sulfasalazine) [164].

In PsO patients who received two doses of the BNT162b2 (Pfizer-BioNTech) mRNA vaccine, humoral response (antibody concentration > 15 U/mL) was comparable to that of healthy controls. However, antibody titers were reduced in patients treated with infliximab and MTX compared with those on biologic monotherapy [166]. In another longitudinal PsO cohort, MTX was also found to impair humoral immune response after one dose of the BNT162b2 (Pfizer-BioNTech) mRNA vaccine, while the response was preserved in patients receiving biologic therapy [167]. However, after two doses of the vaccine, humoral response in patients receiving MTX was comparable to that of healthy controls [168].

A wider breadth of evidence regarding response to SARS-CoV-2 vaccination in IMID patients comes from the IBD field. The ICARUS-IBD study evaluated the serologic response to mRNA vaccines in 48 IBD patients. All IBD patients were seropositive after completing two-dose vaccination and anti-S IgG titers were comparable to those of healthy volunteers. A multiple linear regression analysis showed no association between humoral response and timing of the infusion. They also found reduced immunogenicity in patients on anti-TNF therapy or vedolizumab, although the small sample size and clinical characteristics of the patients may have influenced the result [169]. In contrast, a single-center German cohort study found slightly reduced antibody levels in 72 IBD patients compared with matching healthy controls after two-dose vaccination; however, levels were still considered to be protective. Regarding immunomodulatory therapies, there was no impact on humoral response, but higher antibody titers were observed in patients with a longer interval between the last dose of medication and vaccination [170].

Larger registry studies with improved power to analyze the impact of different immune-modifying drugs are being published. The PREVENT-COVID study is a US prospective, observational cohort study of IBD patients. In an initial report, they noted 95% seropositivity in 317 IBD patients after two doses of mRNA COVID-19 vaccine [171]. In a more recent study of 1909 IBD patients, a positive antibody response within 90 days of the last vaccine dose was observed in 96% of patients receiving mRNA vaccination and in 81% of patients receiving adenovirus vector vaccination. Multivariate analysis found that age, corticosteroids, and anti-TNF in combination with immunomodulators were associated with lack of antibody response. In contrast, patients on vedolizumab or ustekinumab showed a more robust humoral response [172]. Similarly, additional studies have also reported reduced humoral responses in IBD patients treated with TNF inhibitors versus those treated with ustekinumab or vedolizumab [173,174,175], with one study noting more rapid decay in antibody titers in patients receiving TNF inhibitor therapy [173].

The CLARITY IBD study prospectively examined the impact of TNF inhibitor (infliximab) and vedolizumab on SARS-CoV-2 infection and immunity in more than 3000 patients with IBD in the UK. Humoral responses were impaired in infliximab-treated patients following SARS-CoV-2 infection and were further attenuated in those receiving concomitant immunomodulators [186]. After a single dose of either the BNT162b2 (Pfizer-BioNTech) or ChAdOx1-S (AstraZeneca) vaccine, rates of seroconversion (antibody concentrations > 15 U/mL) were lower in patients receiving infliximab than in those receiving vedolizumab, irrespective of the vaccine administered, with only one-third of infliximab-treated patients achieving protective antibody levels [176]. In a more recent report from the CLARITY IBD study analyzing humoral and T-cell responses after a second dose of SARS-CoV-2 vaccine, antibody concentrations were reduced five-fold in infliximab recipients compared with vedolizumab recipients. However, most patients seroconverted after the second dose, with only 6.1 and 1.3% not achieving seroconversion with infliximab and vedolizumab, respectively (*p* < 0.0001). The antibody response was less durable in infliximab-treated patients, with antibody concentrations decaying towards the seroconversion threshold 14–18 weeks after the second dose [177]. Evidence of long-term durability of humoral response is emerging, with several studies reporting reduced durability of humoral response in patients treated with anti-TNF [159,173,174,177].

### 7.2. Cellular Responses to SARS-CoV-2 in IMIDs

Apart from humoral response, cellular immune response is also crucial to generate immune protection against COVID-19. In the CLARITY-IBD study, in which cellular immune response was analyzed in 225 infliximab- and 76 vedolizumab-treated patients, T-cell response was comparable among patients treated with infliximab or vedolizumab, with 20% failing to mount a detectable T-cell response [177]. A smaller study of 28 IBD patients found comparable T-cell response among IBD patients (all receiving immunosuppressive medication) and age- and sex-matched healthy controls [178]. Interestingly, preliminary data from a study analyzing the T-cell clonal response in 303 IBD patients found that antibody and T-cell clonal responses were only modestly correlated, with low T-cell clonal response observed in patients with high antibody levels. The T-cell clonal response was preserved with ustekinumab and vedolizumab and paradoxically augmented by TNF inhibitors [179]. In PsO patients, a lower proportion of patients receiving MTX or biologic therapy showed detectable T-cell response compared with the control group (62 and 71% vs. 100%, respectively) [168]. Moreover, in 51 IMID patients, activated CD8+ T cells were not induced after vaccination in MTX recipients unlike healthy controls or patients on other therapies [165]. In the COVADIS study, which analyzed T-cell response in patients with systemic inflammatory diseases, MTX dramatically impaired T-cell response after two-dose BNT162b2 mRNA vaccination and remained impaired even after the third dose. In contrast, patients on rituximab, despite having poor humoral response, showed T-cell response that were similar to that of controls and further increased after a third dose [187].

### 7.3. Revaccination and New Variants

Indeed, revaccination has been shown to be safe and effective in patients with IMIDs. In a recent study evaluating the effectiveness of SARS-CoV-2 revaccination in IMID patients who failed to respond (seroconversion) to two-dose mRNA vaccination, 80% of patients not previously exposed to rituximab achieved seroconversion after revaccination, whereas only 20% of those treated with B-cell depleting agents seroconverted [188].

Since the start of the pandemic, new SARS-CoV-2 variants have arisen, and some studies have analyzed the effectiveness of vaccination against these new variants in IMID patients. In PsO patients, after two-dose BNT162b2 mRNA vaccination, neutralizing antibody titers against wild-type SARS-CoV-2, Alpha (B.1.1.7), and Delta (B.1.617.2) variants were comparable in patients receiving MTX, biologics, and in healthy controls [168]. In patients with systemic inflammatory diseases, the percentage of patients with neutralizing antibodies against the Delta variant was reduced compared with the Alpha variant for all therapies analyzed [187].

### 7.4. Vaccine Protection against SARS-CoV-2 Infection in IMIDs

The above-mentioned studies evaluated the ability to generate a protective humoral or cellular immune response after SARS-CoV-2 vaccination. However, the most clinically relevant question relates to the ability of these vaccines to prevent infection in patients with IMIDs. An Israeli database study analyzed the real-world effectiveness of two doses of BNT162b2 (Pfizer-BioNTech) vaccine in 12,231 patients with IBD and 36,254 matched patients. They found high vaccine effectiveness with a very low infection rate (0.1%) independent of immune-modifying treatment [180]. Another study evaluating the efficacy of mRNA vaccines against infection in a Veterans Affairs cohort of 14,697 IBD patients in the US (20% were taking TNF inhibitors) found that full vaccination status reduced the risk of infection by 69%, showing that the effectiveness of a two-dose course of an mRNA vaccine was 80.4% in this cohort, which is lower to the efficacy reported in the registration trials [181].

### 7.5. Safety of SARS-CoV-2 Vaccines in IMIDs

SARS-CoV-2 vaccination appears to be safe in patients with IMIDs, with adverse effects comparable to those observed in healthy individuals and no flares of the underlying condition [158,159]. Safety was also studied in PsO patients following mRNA vaccination, with none reporting any adverse effects or a psoriatic flare [182]. Similarly, mRNA vaccines were also safe and well tolerated in a small cohort of PsA patients [164]. In a cohort of 3316 patients with IBD, the PREVENT-COVID registry reported low rates of vaccine-related adverse effects, with only 2% reporting IBD exacerbations following vaccination [183]. In addition, preliminary data from the Corale-IBD registry suggest that the risk of vaccine-related adverse events is not increased after vaccination with mRNA vaccines [184] and that post-vaccination symptoms after the third dose were similar, milder, and less frequent than those after a second dose [185].

### 7.6. Guidelines on SARS-CoV-2 Vaccination in IMIDs

Guidelines on SARS-CoV-2 vaccination in patients with IMIDs are beginning to emerge from major international organizations, including the US National Psoriasis Foundation (NPF) [22], American Academy of Dermatology [189], American College of Rheumatology (ACR) [115], European Academy of Dermatology and Venereology [190], British Society of Gastroenterology [191], and the International Organization for the Study of IBD [25]. All recommend SARS-CoV-2 vaccination for patients with IMIDs, including those on biologic therapy [22,25,115,189,190,191,192]. Most of these guidelines were released prior to any data being available and recommend continuing biologic therapy in patients receiving SARS-CoV-2 vaccines. However, the ACR guidelines recommend withholding MTX and JAK inhibitors for 1 week after each vaccine dose and withholding abatacept for 1 week before and 1 week after the first vaccine dose. They also recommend delaying rituximab for 2–4 weeks after the second vaccine dose if the patient’s disease activity allows [115]. The lower rate of seroconversion after one dose highlights the importance of the timing of the second dose in immunocompromised patients. Moreover, the rapid decline in antibody levels after the second dose, the emergence of new variants such as Omicron, and preliminary data showing the benefits of a third dose, underline the importance of a third primary dose for individuals who are immunocompromised (i.e., those with immunocompromising conditions or receiving immunosuppressive therapy, which includes many IMID patients), as it is now recommended by the European Medicines Agency [193], the UK Joint Committee on Vaccination and Immunisation [194], and the US Centers for Disease Control and Prevention [195].

### 7.7. Evidence Summary on SARS-CoV-2 Vaccination in IMIDs

SARS-CoV-2 vaccination in IMID patients treated with biologic therapies appears to be safe and well tolerated. Under most immune-modifying therapies, humoral and cellular immune responses were preserved, except for B-cell depleting agents, where humoral immune response was impaired, MTX which impaired humoral and cellular response, and TNF inhibitors which blunted humoral immunity.

## 8. Conclusions

Patients with IMIDs may be at increased risk of developing vaccine-preventable infections, and therefore vaccination strategies are an important preventive measure in these patients. Despite blunted immunogenicity observed with some agents, vaccination is effective and safe and therefore patients and physicians should not be dissuaded from vaccination. The greatest reduction in vaccine response has been observed in patients taking B cell-depleting agents, followed by TNF inhibitors, and the CTLA-4 fusion protein abatacept; however, no attenuation has been noted with anti-integrin α4β7, anti-BAFF, anti-IL-6, anti-IL-17, or anti-IL-12/23 agents. Data on the novel SARS-CoV-2 vaccines in IMID patients are rapidly accumulating and seem to coincide with what is known about other vaccines. In general, SARS-CoV-2 vaccines are safe and effective in most IMID patients, although they may benefit from a third dose as currently indicated.

## Figures and Tables

**Figure 1 vaccines-10-00297-f001:**
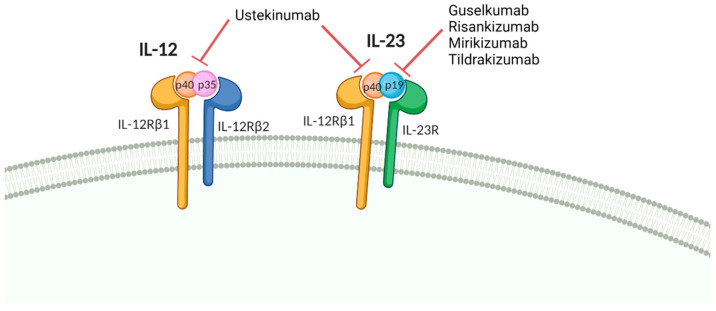
Schematic representation of IL-12 and IL-23 cytokines and its receptors. Figure created with Biorender.com [141].

**Table 1 vaccines-10-00297-t001:** Effect of biologic therapies on response to vaccination in IMID patients.

Biologic Therapy	Vaccine	Population	Impact on Vaccine Response
**Integrin inhibitors**			
Natalizumab (anti-α_4_β_1_)	Influenza	MS	Reduced humoral response [47,48]
	Tetanus	MS	Normal humoral response [49]
Vedolizumab (anti-α_4_β_7_)	Influenza	IBD	Normal humoral response [50]
	Oral cholera	Healthy volunteers	Reduced humoral response [51]
	Hepatitis B	Healthy volunteers	Normal humoral response [51]
**CTLA-4 fusion protein**			
Abatacept	PPV-23	RA	Reduced antibody titers, but normal functional antibodies [52]
			Normal humoral response [53]
	PCV-7	RA	Impaired humoral response [54]
	Influenza	RA	Normal humoral response [53]
			Impaired humoral response [55]
**CD20 inhibitor**			
Rituximab	PPV-23	RA	Impaired humoral response [56]
		IMIDs	Impaired humoral response [57]
	PCV-7	RA	Impaired humoral response [54]
		IMIDs	Impaired humoral response [57]
	Hepatitis B	IMIDs	Impaired humoral response [58]
	Tetanus	RA	Preserved humoral response to recall antigen [56]
	Influenza	RA	Impaired humoral [59,60,61,62], but preserved cellular response [59]
		IMIDs	Impaired humoral response [63]
		RA or vasculitis	Impaired humoral response [64,65]
**BAFF inhibitor**			
Belimumab	PCV-13	SLE	Normal humoral response [66]
	PPV-23	SLE	Normal humoral response [67]
**TNF inhibitors (aggregated)**			
	PPV-23	SpA	Reduced humoral response [68]
		PsA	Normal humoral response [69]
		RA	Reduced humoral response [70]
			Normal humoral response [71,72,73]
		IBD	Reduced humoral response [74,75,76,77]
	PCV-13 then PPV-23	IBD	Impaired humoral response [77]
	PCV-13	RA	Impaired humoral response [78]
		CD	Impaired humoral response [74]
	Hepatitis B	SpA	Reduced humoral response [68]
		IBD	Impaired humoral response [79,80], or seroconversion rates similar to thiopurine or MTX [81]
	Hepatitis A	RA	Impaired humoral response [82]
	Influenza	RA	Normal humoral response [71,72,83,84,85,86,87]
		AS	Normal humoral response [83]
		SpA	Reduced humoral response [85]
		PsA or PsO	Normal humoral response [88]
		IBD	Reduced humoral response [89,90,91,92]
		IMIDs	Reduced humoral response [93,94]
	MMR	JIA	Normal humoral and cellular response [95]
	Yellow fever	RA	Preserved antibody response to revaccination [96]
**IL-17 inhibitors**			
Secukinumab	Meningococcal conjugate	Healthy volunteers	Normal humoral response [97]
	Influenza	Healthy volunteers	Normal humoral response [97]
		PsA or AS	Normal humoral response [98]
Ixekizumab	PPV-23	Healthy volunteers	Normal humoral response [99]
	Tetanus	Healthy volunteers	Normal humoral response [99]
**IL-6 inhibitor**			
Tocilizumab	PPV-23	RA	Normal humoral response [100,101,102]
	PCV-13 then PPV-23	IMIDs	Normal humoral response [57]
	PCV-7	RA	Normal humoral response [54]
	Influenza	RA	Normal humoral response [64,102,103]
		SJIA	Normal humoral response [104]
	Tetanus	RA	Normal humoral response [100]
**IL-12/IL-23 inhibitors**			
Ustekinumab	PPV-23	PsO	Normal humoral response [105]
		PsO or MS	Normal humoral response [106]
	Influenza	CD	Normal humoral response [107]
	Tetanus	PsO or MS	Normal humoral response [106]
		PsO	Normal humoral response [105]

AS, ankylosing spondylitis; BAFF, B-cell activating factor; CD, Crohn’s disease; CTLA-4, cytotoxic T-lymphocyte-associated protein 4; IBD, inflammatory bowel disease; IL, interleukin; MS, multiple sclerosis; IMID, immune-mediated inflammatory diseases; JIA, juvenile idiopathic arthritis; MMR, measles-mumps-rubella; MS, multiple sclerosis; NSAID, non-steroidal anti-inflammatory drugs; PCV-7 or PCV-13, 7-valent or 13-valent pneumococcal conjugate vaccine; PPV-23, 23-valent pneumococcal polysaccharide vaccine; PsA, psoriatic arthritis; PsO, psoriasis; RA, rheumatoid arthritis; SJIA, systemic juvenile idiopathic arthritis; SLE, systemic lupus erythematosus; SpA, spondyloarthropathy; TNF, tumor necrosis factor.

**Table 2 vaccines-10-00297-t002:** Graphical summary on the effect of biologic therapies on response to vaccination in IMID patients.

Biologic Therapy	Impact on Vaccine Response ^a^	
CD20^+^ cell depletion		
CTLA-4 fusion protein		
TNF inhibitors		
Integrin inhibitors		Oral cholera
BAFF inhibitor		
IL-17 inhibitors		
IL-6 inhibitor		
IL-12/IL-23 inhibitors		

^a^ Red shading indicates impaired humoral response, orange indicates reduced humoral response, and green indicates normal humoral response. BAFF, B-cell activating factor; CTLA-4, cytotoxic T-lymphocyte-associated protein 4; IL, interleukin; TNF, tumor necrosis factor.

**Table 3 vaccines-10-00297-t003:** Immunogenicity and safety of SARS-CoV-2 vaccines in IMIDs.

SARS-CoV-2 Vaccine	Patient Population	Biologic Therapy	Effect on Immunogenicity	Safety	Reference
mRNA-1273 and BNT162b2	CID (*n* = 133)	29% receiving anti- TNF, 9% anti-integrin, 8% anti-CD20, 8% anti-IL-12/23 or anti-IL-23, 2% anti-BAFF, 2% CTLA-4, and 1% each anti-IL-6 and anti-IL-1	Corticosteroids and B cell-depleting therapies strongly impaired humoral response. JAKi and antimetabolites (e.g., MTX) blunted humoral responses. Anti- TNF, UST, and VDZ had minimal impact	Not reported	Deepak et al. 2021 [157]
BNT162b2	IMID (*n* = 84)	13% receiving anti-TNF, 8% anti-IL-17, 7% anti-IL-23, 4% anti-IL-6, 1% anti-IL-1, 1% anti-integrin	Impaired humoral response compared with healthy controls independent of treatment	AE similar to general population	Simon et al. 2021 [158]
mRNA-1273 and BNT162b2	CID (*n* = 26)	50% receiving anti-TNF, 12% anti-IL-17, and 4% each anti-IL-6, anti-IL-12/23, and anti-integrin	Slightly reduced humoral response	AE comparable to general population and no flares of CID	Geisen et al. 2021 [159]
BNT162b2	IRD (*n* = 264)	24% receiving anti-TNF, 18% anti-CD20, 15% anti-interleukins, and 3% CTLA-4	Significant humoral response in majority of patients, except those receiving RTX	Minor adverse effects and no flares of IRD	Braun-Moscovici et al. 2021 [160]
mRNA-1273 and BNT162b2	RA (*n* = 53)	47% receiving biologic therapy	Significantly lower antibody response in patients with RA vs. healthy individuals. Lowest rate of response in patients receiving JAKi	Not reported	Rubbert-Roth et al. 2021 [161]
BNT162b2	RA (*n* = 83) PsA (*n* = 29) SpA (*n* = 28)	44% receiving anti-TNF, 21% CTLA-4, 14% anti-IL-12/23, and 10% anti-IL-6	Impaired immunogenicity after one dose in patients treated with MTX, glucocorticoids, and abatacept. No effect of cytokine inhibitors	Not reported	Bugatti et al. 2021 [162]
BNT162b2 and ChAdOx1-S	IMID (*n* = 120), mostly PsO	83% receiving biologic therapy	Impaired immunogenicity after one dose in patients treated with MTX compared to biologics	Not reported	Al-Janabi et al. 2021 [163]
BNT162b2	PsA (*n* = 40)	100% receiving anti-TNF	Anti-TNF numerically (not significantly) decreased humoral response vs. healthy controls; MTX, glucocorticoids, and sulfasalazine had no impact	AE similar to general population; no changes in clinical disease activity	Venerito et al. 2022 [164]
BNT162b2	IMID (*n* = 51) mostly PsO and/or PsA (*n* = 24) RA (*n* = 22)	55% receiving biologic therapy	Impaired humoral and cellular responses in patients receiving MTX vs. IMID patients on other DMARDS or healthy controls	Not reported	Haberman et al. 2021 [165]
BNT162b2	PsO (*n* = 48)	44% receiving anti-TNF, 27% anti-IL-23, 17% anti-IL-12/23, 13% anti-IL-17	Reduced humoral response in patients receiving biologics in combotherapy vs. monotherapy	No increase in AEs and no flares	Cristaudo et al. 2021 [166]
BNT162b2	PsO (*n* = 84)	80% receiving biologic therapy	Impaired humoral response with MTX after first dose but preserved with biologics; after 2 doses, responses comparable to controls	Not reported	Mahil et al. 2021, 2022 [167,168]
mRNA-1273 and BNT162b2	IBD (*n* = 48)	33% receiving anti-TNF, 42% anti-integrin, 8% anti-IL-12/23, and 2% anti-IL-23	100% seropositivity after 2 doses but reduced serologic response in patients on anti-TNF or VDZ	Not reported	Wong et al. 2021 [169]
mRNA-1273, BNT162b2 and ChAdOx1-S	IBD (*n* = 72) Healthy controls (*n* = 72)	37% receiving anti-TNF, 26% anti-integrin, and 19% anti-IL-12/23	Slightly decreased antibody titers in IBD patients compared to healthy controls	Well tolerated with only mild side effects	Classen et al. 2021 [170]
mRNA-1273 and BNT162b2	IBD (*n* = 317)	42% receiving anti-TNF, 15% anti-integrin, and 12% anti-IL-12/23	95% had detectable antibodies	Not reported	Kappelman et al. 2021a [171]
mRNA-1273, BNT162b2 and Ad26.COV2.S	IBD (*n* = 1909)	47% receiving anti-TNF, 15% anti-IL-12/23, and 12% anti-integrin	96% achieved positive antibody response; age, corticosteroids, and anti-TNF + IMM associated with reduced odds of antibody response	Not reported	Kappelman et al. 2021b [172]
mRNA-1273 and BNT162b2	IBD (*n* = 176)	27% receiving anti-TNF, 27% anti-IL-12/23, and 11% anti-integrin	Significantly lower antibody titers and more rapid decay with anti-TNF ± IMM vs. UST, VDZ, or no therapy	Not reported	Charilaou et al. 2021 [173]
mRNA-1273 and BNT162b2	IBD (*n* = 75)	51% receiving anti-TNF, 23% anti-IL-12/23, and 8% anti-integrin	Detectable antibodies 6 months after 2-dose vaccination; patients receiving anti-TNF had lower antibody titers	Not reported	Frey et al. 2022 [174]
BNT162b2 and ChAdOx1-S	IBD (*n* = 126)	76% receiving anti-TNF, 12% anti-IL-12/23, and 12% anti-integrin	Majority of patients receiving anti-TNF and VDZ and all on UST seroconverted; neutralizing antibody concentrations were higher with UST and VDZ	Not reported	Shehab et al. 2021 [175]
BNT162b2 and ChAdOx1-S	IBD (*n* = 2977)	69% receiving anti-TNF and 31% anti-integrin	Five-fold reduction in humoral response with IFX vs. VDZ; more rapid decay in antibody levels in IFX-treated patients	Not reported	Kennedy et al. 2021 [176] Lin et al. 2021 [177]
BNT162b2 and ChAdOx1-S	IBD (*n* = 28) Healthy controls (*n* = 27)	32% receiving anti-TNF, 29% anti-IL-12/23, 11% anti-integrin	Comparable T-cell responses between IBD patients and healthy controls	Not reported	Reuken et al. 2021 [178]
mRNA-1273, BNT162b2 and Ad26.COV2.S	IBD (*n* = 303)	35% receiving anti-TNF, 43% anti-IL-12/23 or anti-integrin	Preserved T-cell clonal response with UST and VDZ and augmented by anti-TNF	Not reported	Li et al. 2021 [179]
BNT162b2	IBD (*n* = 12,231) and matched patients (*n* = 36,254)	11% receiving anti-TNF, 4% anti-integrin, and 2% anti-IL-12/23	Similar risk of infection between controls and IBD patients, with no effect of immune-modifying therapies	Not reported	Ben-Tov et al. 2021 [180]
mRNA-1273 and BNT162b2	IBD (*n* = 14,697)	Not reported	Two-dose vaccination reduced the hazard of infection by 69%, with an estimated efficacy of 80.4%	Not reported	Khan et al. 2021 [181]
mRNA-1273 and BNT162b2	PsO (*n* = 50)	100% receiving biologic therapy	Not reported	AE similar to general population	Musumeci et al. 2021 [182]
mRNA-1273, BNT162b2 and Ad26.COV2.S	IBD (*n* = 3316)	46% receiving anti-TNF, 15% anti-IL-12/23, and 12% anti-integrin	Not reported	Low rates (2%) of IBD flare following vaccination and relatively few vaccine-related AEs	Weaver et al. 2021 [183]
mRNA-1273 and BNT162b2	IBD (*n* = 246)	37% receiving anti-TNF, 17% anti-IL-12/23, and 13% anti-integrin	Not reported	AE similar to general population	Botwin et al. 2021 [184]
mRNA-1273 and BNT162b2	IBD (*n* = 524)	89% receiving biologic therapy	Not reported	Post-vaccination symptoms after third dose are generally milder and less frequent than after second dose	Li et al. 2021 [185]

AE, adverse event; CID, chronic inflammatory diseases; CTLA-4, cytotoxic T-lymphocyte-associated protein 4 fusion protein; IBD, inflammatory bowel disease; IFX, infliximab; IL, interleukin; IMID, immune-mediated inflammatory diseases; IMM, immunomodulator; IRD, inflammatory rheumatic diseases; JAKi, Janus kinase inhibitors; MTX, methotrexate; PsA, psoriatic arthritis; PsO, psoriasis; RA, rheumatoid arthritis; RTX, rituximab; SpA, spondyloarthritis; TNF, tumor necrosis factor; UST, ustekinumab; VDZ, vedolizumab. mRNA-1273 (Moderna, Cambridge, US), BNT162b2 (Pfizer-BioNTech, NYC, US), Ad26.COV2.S (Janssen, Beerse, Belgium), and ChAdOx1-S (AstraZeneca, Cambridge, UK).

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
