# Peer review of "Response to Vaccines in Patients with Immune-Mediated Inflammatory Diseases: A Narrative Review"

_vaccines, 2022, doi:10.3390/vaccines10020297_

Round 1
Reviewer 1 Report
In this review, Garcillan et al present an overview of the efficacy of a variety of vaccines in individuals who suffer from IMIDs (with particular emphasis on those who are taking biologics to treat their conditions). The authors also dedicate a section of their review describing studies of Covid-vaccine efficacy in patients with IMIDs. The review is extremely well-written, and the reviewer would be satisfied with the acceptance of the review in its current form.
Reviewer 2 Report
I would like to congratulate the authors. This article is a great effort and there is no doubt about the need to compile the existing information on the subject of the present manuscript.
I propose to accept the article after minor revision, specifically to include the incipient experience (if any) after the third dose and to adapt the language to the current situation in which the booster dose is an established recommendation. That would be an important update considering that the question of the booster dose has been established as the state of the art today due to the emergence of Omicron, and even before because of Delta.
Similarly, and considering that the protection levels are changing after each new variants, it would be wise to name the name of the variant after referring to any specific level of protection of the induced antibodies (if it is possible to name or alternatively to suggest), although I recognize that sometimes it won't be possible.
Author Response
Please, see the attachment

Reviewer 3 Report
Comments to the Authors
The authors presented an extensive review on the topic of vaccination response in patients with IMID treated with biologics.
Would recommend using the uniform terminology through the whole review.
For instance, in the abstract “biologic agents”, biologic drugs”, biologic therapies” are interchangeably used. Please consider to use the same terminology/definitions.
Abstract:
“Biologic drugs used to treat IMIDs target 14 the immune system to stop pathogenic chronic inflammation” - please change to chronic pathogenic process.
“However, and despite limited data, the efficacy of 25 SARS-CoV-2 vaccines on IMID patients may be reduced compared with healthy individuals.” Please remove “ and despite limited data”
Line 28: What does ”highly immunosuppressive drugs” mean? Either explain or remove the word “highly”.
Introduction
“The aim of this narrative review is to summarize current knowledge regarding vaccine responses in IMID patients treated with biologics, with a focus on agents that inhibit interleukin (IL)-12 and/or -23.” Please consider to explain the rationale for expanding the data on this particular biologic group.
According to the abstract: “Here, we review the current knowledge on the impact of biological therapies on immune response to vaccines in IMID patients.” Please unify the aim of the review. If there is a focus on the impact of IL-12/23 inhibitors, this should also be stated in the abstract. In addition, please define the scope of the IMID included in this review.
Section 2. Line 75
“The risk of most infections is lower during treatment with IL-12/23 or IL-17 inhibitors compared with TNF inhibitors [19]” The reference 19 refers to patients with psoriasis/PsA. Another study entitled “Comparative risk of serious infections among real-world users of biologics for psoriasis or psoriatic arthritis” by Li et al states “After adjustment for propensity scores, there was no increased risk with IL-17 compared with either TNF (HR=0.89, 95% CI 0.48 to 1.66) or IL-12/23 (HR=1.12, 95% CI 0.62 to 2.03). In general, would suggest to avoid the comparison of the infection risk between biologic agents but only in case of a consensus statement. If decided to keep the comparison, please expand in which diseases/patients’ population this observation was made. In my opinion, such a comparison is out of the scope of this review.
Line 88:
“Biologic treatments were 88 not associated with an increased risk of developing severe COVID-19”.
This is not a precise information as some studies pointed out that patients treated with rituximab had a severe COVID-19 disease course and poor outcome.
Lines 107-108:
“although some studies 107 reported increased joint pain following the administration of some vaccines in patients 108 with rheumatologic conditions treated with TNF inhibitors or MTX” Please omit the last part of the sentence (underlined in bold) as irrelevant.
Section 7. The data on COVID-19 vaccines is rapidly accumulating. This section should be updated with the most recent literature in field.
